# Research on the Impact of Popular Tourism Program Involvement on Rural Tourism Image, Familiarity, Motivation and Willingness

**Min-Yuan Zheng [1], Chao-Chien Chen [2], Hsiao-Hsien Lin [3]****, Chih-Hung Tseng [2] and Chin-Hsien Hsu [3],***

[1] Wuhan Institute of Technology, School of Art & Design, Wuhan 430205, China; zhengminyuan0623@gmail.com
[2] Department of Leisure and Recreation Management, Asia University, Taichung 41354, Taiwan; peter72@asia.edu.tw (C.-C.C.); boy217010@hotmail.com (C.-H.T.)
[3] Department of Leisure Industry Management, National Chin-Yi University of Technology, Taichung 41170, Taiwan; chrishome12001@yahoo.com.tw
* Correspondence: hsu6292000@yahoo.com.tw

**Abstract:** The purpose of this study was to investigate the influence of people's involvement in tourism programs on the imagination and familiarity of rural tourism destinations, as well as the motivation and desire to visit them. The results were validated by using SPSS 22.0 and AMOS 21.0 statistical software to calculate and analyze the samples and to explore and infer the causal relationships. The results showed that: the degree of travel program involvement significantly correlated with familiarity and imagery of the destination ($p < 0.000$), and the degrees of familiarity and imagery of the destination also significantly correlated with motivation to travel ($p < 0.000$), and, lastly, the higher the motivation to travel in the last village, the greater the desire to travel ($p < 0.000$).

**Keywords:** environment; tourism marketing; social media; travel program; degree of involvement

## 1. Introduction

Tourism has emerged following the advancement of human technology and the improvement of quality of life. With the active development of the tourism industry and activities in various countries, the tourism industry has become a sunshine industry actively developed by various countries around the world. It has become a closely integrated activity in the life of human beings. The tourism industry is a human behavior that is formed by attracting people from abroad to travel and spend money through the natural environment, landscape, human customs, food, and the special environment of a place [1]. It has become one of the major economic industries that countries are striving to develop and promote because of its low cost, fast development speed, and high revenue effect [2].

The economic benefits associated with tourism development are evident. In order to achieve greater benefits, each country has been exploring local characteristics, promoting micro-tourism and in-depth tourism activities, and assisting in promoting various marketing strategies by proactively organizing marketing departments or commissioning advertising programs and utilizing media such as online media and print media for marketing. Through the marketing tactics of outsider and celebrity hosts, food programs, food photo clips, videos, etc., existing tourism strategies can be invigorated [3]. A new form of tourism marketing is emerging in which program topics are given to media organizations or studios employing celebrity or outsider presenters to produce or distribute travel programs [4].

The marketing of tourism by media platforms has become a means and trend of marketing tourist destinations and products. With the development of media technology, the speed of internet communication and the number of visitors, and the rise of the information explosion generation, tourism programs can be produced in the most realistic

and rapid way to present the characteristics of local tourism [5]. Besides, the COVID-19 epidemic has caused a significant decrease in tourists' willingness to travel, and the global tourism industry is in recession [6]. Travel programs delivered through real-time video, experienced and narrated by real people, have become one of the main ways for people to receive and understand the imagery and activities of tourist destinations.

Tourism video production intends to provide the public with first-hand tourism information of developed or to-be-developed tourist destinations such as natural scenic spots or villages through the introduction of the video, using the scenery filmed in the field, explanations by experts, and perceptions and opinions after personal experience. This approach has been successful in attracting domestic and foreign tourists, effectively increasing tourism flows in Taiwan, China, Japan, Thailand, and other regions [7,8], and indirectly promoting the development of tourism media-related industries, resulting in profits for producers and hosts [9,10], and creating business opportunities for tourism industries around the world. It is clear that tourism programs not only help governments to successfully promote tourism development strategies, but also drive the development of tourism media, local villages, and tourism-related industries, and quickly raise the public's awareness of the tourism style, resources, and characteristics of tourist destinations. As a result, if people want to plan travel activities in their leisure time, tourism programs may become a key factor in their planning of activities and their willingness to engage in travel.

By presenting videos or explanations, tourism programs use a variety of editing and filming techniques to attract public recognition of the programs, increase viewership, and enhance program popularity. Because of their high exposure rate, high-profile tourism programs are the best medium for local governments, businesses, and the public to advertise and market. It has become a major trend for travel program operators to provide local material and program organizations to edit content to help promote and market their products. In addition, travel programs can record real-time information and images, and broadcast them indefinitely to disseminate information, raise the awareness of tourist destinations and businesses, and increase people's knowledge of tourist destinations in order to promote tourism motivation and ultimately increase travel desire, stimulate consumption, and create a profitable situation for multiple parties. Sustainable rural tourism is a long-term goal, which is to create a close relationship between tourism industry development and the natural environment, cultural identity, and community participation in rural areas [11]. This is the ultimate goal of tourism programs and the expected contribution to the development of tourism regions and local industries.

Travel intention is influenced by multiple dimensions. Lack of instant and authentic information is the main key to influence the desire to travel [12]. Tourism programs are the best way for people to get instant and actual tourism information and the most realistic feelings about the tourist destinations. It is a type of media communication that uses local tourism resources as materials, exploits characteristics such as natural scenery or human customs, and produces attractive program content to attract the public's attention, boosts program ratings for tourism program organizations, corporate organizations, or individual studios, and assists local governments, corporations, organizations, businesses, or individuals in marketing and promoting tourism industries such as tourism commodities, in order to increase the popularity of tourist areas, attract people, promote the development of tourism industries, and advance the local economy and the program producers' own profits [13]. The public can obtain correct information and reduce travel losses through program content planning and video explanations [8,9]. However, tourism information will change with various factors such as tourism policies, technological advancement, corporate investment or relocation, and climate change, resulting in seasonal or immediate differences in the current conditions of tourism destinations [14]. This suggests that it takes four transitions in the travel experience before researchers begin to believe that the experience is the result of a person's interpretation of the culture, time period, and context of the visit, and this view is similar to the interactive experience model proposed by Falk and Dierking [15].

Therefore, receiving real-time information and updating information effectively and frequently will be the best way for tourists to reduce travel mistakes.

Although tourism programs are designed and planned with local tourism resources as the subject matter, they are designed to provide information for the public to watch and take in, but due to the rapid changes in technology, information is constantly evolving. If the public has the wrong level of involvement in the information of tourism programs, it may affect the public's perception of the current situation of tourism destinations and create misunderstandings about the contents of local tourism products and activities, resulting in misconceptions about the imagery and knowledge of tourism destinations, which will eventually affect the public's willingness to consume in tourism destinations. Because destination imagery is the key to tourists' perceptions of the destination [16], it is the emotional link between the consumer's psychology and the tourist destination, and plays a role in the overall attractiveness of the viewing system, generating positive, neutral, and negative feelings [17]. The current status of the imagery of the tourism destination affects the tourists' motivation to travel and therefore deters them from doing so, affecting their behavioral intention [18].

Travel familiarity refers to consumers' product awareness and related experience with products, services, consumption, and information [19,20]. It is the degree of familiarity with a destination that results from the systematic reorganization and arrangement of information and experiences received from various channels or past visits to the destination [21]. As individuals accumulate more experience and familiarity with tourism products, they change tourists' perceptions of destinations and develop the ability to analyze and interpret information [22], which influences tourists' willingness to travel. It is evident that destination imagery and familiarity both play a significant role in tourists' motivation to travel.

By exploring the factors that influence travel motivation, we can ultimately understand people's desire to travel. Travel motivation is an expression of the various key factors that lead individuals to engage in travel behavior and can present the main reasons why tourists engage in travel activities [23,24]. Moreover, travel motivation can be seen as the internal force that drives tourists to participate in a particular tour, prompting people to seek out travel activities that meet their needs and also guiding them to conduct travel activities to satisfy their needs and purposes [25]. This result of multiple assessments of internal personal influences on tourism, which can be seen as the expectation and intention to perform a trip, is a result of multiple assessments of internal personal influences on tourism and is a key influencing factor in deciding whether or not to perform a tourism behavior [26].

As mentioned above, it is the goal of local governments and enterprises in tourist areas to increase the public's positive imagery of tourist destinations, enhance familiarity with local industries, products and activities, and induce the motivation and desire to travel. It is also the original goal of tourism program producers to help local governments and enterprises in tourism areas to effectively market and successfully promote the prosperity of tourism areas and raise local awareness. Travel programs have the potential to influence travel intentions, but differences in the degree of involvement in travel programs affect people's imagery and familiarity with travel destinations. The difference in the level of familiarity and imagery of travel destinations affects people's motivation to travel, and the difference in people's motivation to travel leads to the final result of travel intention. Therefore, the study suggests that by analyzing people's level of involvement in tourism programs by first understanding the influence of differences in destination imagery and familiarity generated by different levels of involvement on travel motivation, and then exploring whether travel motivation influences the outcome of travel desire, the study will help to understand the influence of tourism programs on tourism development and tourists' participation in tourism.

Furthermore, in order to construct a better and more rigorous theoretical foundation, a review of the survey revealed that there is a wide range of studies that individually explore

the issues of tourism involvement, tourism imagery, tourism motivation, and tourism intention, and there is a considerable amount of research. A considerable number of people have explored the extent of travel program involvement on travel imagery [27,28] and travel motivation on travel intention [29]. However, there are few studies on the degree of involvement in similar travel programs or online information [30]. There has been no research conducted on the logic and issues of analyzing the differences in tourism destination imagery and familiarity generated by people's involvement in tourism programs, then exploring the effects of the differences in tourism destination imagery and familiarity on tourism motivation, and further understanding the changes in tourism motivation due to the differences in tourism destination imagery and familiarity generated by the degree of involvement in programs, and whether they have changed the tourism intention. Therefore, the researchers believed that it would be helpful to understand the influence of travel programs on people's travel intention by taking the level of involvement in travel programs as the main axis, understanding the differences in travel destination imagery and familiarity at different levels of involvement, and the relationship between travel destination imagery and familiarity on travel motivation, and then exploring their influence on travel motivation and travel intentions.

## 2. Literature Review

### 2.1. Perception Analysis of People's Travel Intention

Consumers decide whether or not to purchase a product by referring to information about the product to satisfy a certain need and then make a judgment based on the information or their internal and external needs [31,32]. Tourism is a commodity, and the process and behavior of tourists to participate in tourism activities or purchase goods in tourist areas to satisfy their individual internal and external needs to achieve various tourist purposes is called the travel intention.

Travel intention has an influence on tourists' spending in tourist destinations; the higher the tourists' travel intentions, the higher their willingness to travel and spend in the destination [33]. Conversely, when tourists have negative emotions or perceptions, they are less likely to have travel intentions [34,35]. Moreover, the intention to travel varies among consumers from different backgrounds, including gender, age, occupation, education, family status, and monthly income [36,37].

Based on the above literature and descriptions, the researchers believed that people with different backgrounds such as gender, age, occupation, education level, family status, and monthly income may have different travel intentions after watching travel programs due to the differences in their background characteristics, and therefore this issue needs to be further examined.

### 2.2. Correlation between the Degree of Travel Program Involvement and Destination Imagery

Destination imagery is a key factor in tourists' behavior during the travel decision process and destination selection [38]. When tourists have a positive image of a destination, it will encourage them to visit it [39,40]. However, the tourism resources of tourist destinations are diverse, and in addition to the developed tourist attractions, there is a wealth of tourism features that have yet to be explored, and without personal experience and understanding, much of the knowledge and joy that comes with tourism will be missed.

The tourism program makes use of the existing local resources and presents the dazzling tourism characteristics of the local area through a single video or explanation, or even a mix of them so that the public can have a deeper understanding of the local characteristics. The more information that is provided and the more frequent it is, the more impressive it is to the public. This mode of information provision is an alternative way to compensate for the fact that people are unable to experience the place in person and thus miss out on receiving travel information. Studies have shown that the degree to which people have access to travel information affects their perceptions and impressions of

a destination. Good destination imagery can facilitate travel planning [41–43], while the opposite may affect tourists' willingness to travel.

Therefore, researchers believed that travel programs do help people to understand the information of the place they are visiting. The frequency of transmission or the number of times it is received may influence people's desire to travel to a destination. High intention increases positive perceptions of the tourist area and vice versa [43]. Therefore, the researcher believed that the study of people's destination imagery after being involved in watching tourism programs can help to understand the influence of tourism programs on tourists' destination imagery, and subsequently to understand whether tourism programs are a key factor influencing people's desire to travel. Therefore, the researcher believed that the relationship between the level of involvement in travel programs and destination imagery needs to be further examined.

### 2.3. The Correlation between the Degree of Travel Program Involvement and Travel Familiarity

Familiarity can be considered as the degree of product awareness and the related experience of products, services, consumption, and information accumulated by consumers [19]. In the case of tourism, familiarity refers to tourists' understanding and knowledge of the place they are visiting. Usually, the general public is unfamiliar with the sights, characteristics, culture, and products of a tourist destination without experiencing them firsthand or learning to receive information on their own. The production, planning, and broadcasting of tourism programs can meet the demand for resources to understand the scenery, characteristics, culture, and products of tourist areas. By adjusting the duration and frequency of the allocation, a more complete amount of tourism information can be transmitted. The more sufficient tourism information is available, the more it can compensate for the lack of information about the tourist area, and the more it can improve the positive perception of the tourist area [44,45].

Therefore, the researchers believed that tourism programs do help to increase people's knowledge of the places they visit and that the frequency of transmission or the number of times they are received will be crucial in influencing people's desire to travel to tourist areas. Therefore, the researchers believed that it is necessary to examine the relationship between the degrees of involvement in tourism programs and travel familiarity.

### 2.4. Correlation between Destination Imagery and Travel Motivation

Motivation induces the behavior that people are about to enact [46]. Travel motivation is an invisible influence that affects people's imminent travel behavior and directs that behavior towards a goal that may actually be accomplished. Usually, the motivation can be induced by fitness, play, entertainment, recreation, creativity, and relaxation activities to achieve physical and mental health and stress relief [47]. In the case of tourism, people may be attracted to travel and spend money due to various tourism resources that are political, economic, social, folkloric, architectural, literary and artistic, or environmental [48]. The more people know about the tourist places, the quicker they can plan for future tourism. Therefore, the more people are involved in tourism programs, the more opportunities they have to obtain more and better tourism information, the higher the positive imagery and impression of the destination [49], and the greater the desire to travel and spend money there.

Therefore, the researchers believed that tourism programs do help to increase people's knowledge of tourist destinations and can change people's perceptions of tourist destinations by enhancing destination imagery, which can lead to the desire to travel and spend money. Therefore, the researchers believed that it is necessary to examine the relationship between destination imagery and travel motivation.

### 2.5. Correlation between Travel Familiarity and Travel Motivation

Travel motivation is a psychological expectation that causes people to travel and spend money after collecting, reading, analyzing, and judging various pieces of travel

information [50]. In order to construct a perfect and ideal tourism plan, people themselves must know the tourist places. Tourism programs are designed to stimulate the interest of the viewer and change the way he or she sees and enjoys the scenery of the destination, thus attracting consumers to the destination [50,51]. The more distinctive and impressive the presentation of the destination, the greater the desire to travel to it [52].

Therefore, the researchers believed that the increase in knowledge of the tourist destination and the understanding of the local tourism resources as a result of the tourism program may change people's perception of the tourist destination and lead to the desire to travel and spend money there. Therefore, the researcher believed that it is necessary to examine the relationship between travel knowledge and travel motivation.

### 2.6. Correlation between Travel Motivation and Travel Intention

Travel motivation is a key factor in determining whether or not to perform travel behavior [53], as it can be considered as an assessment of the multiple factors that influence travel within an individual's mind. It influences people's perceptions of tourist places through past travel experiences or acquired travel information, and further constitutes tourists' attitudes and induces travel motivation [54], thus creating a prototype of travel decision-making with scale, system, and method. This can usually be explored in terms of internal and external factors and by the interplay of push and pull forces [55]. Travel programs can provide sufficient travel information to satisfy people's internal emotional power [56], personal needs, and desires, external travel information about environmental features or characteristics, and the stronger the attraction and higher the motivation generated, the more likely it is to change or influence people's travel intentions [53].

Therefore, the researchers believed that the motivation to travel would be strengthened by the fact that tourism programs increase people's imagery and knowledge of the tourist destinations, which may influence people's desire to travel. Therefore, the researchers believed that it is necessary to examine the relationship between travel motivation and travel intentions.

## 3. Methodology

### 3.1. Research Process

The present study aimed to investigate the degree of people's involvement in watching travel programs and their imagery, familiarity, motivation, and intention to travel to tourist destinations. The research process involved first reading the literature to clarify the main directions of the study, then integrating the findings of the literature, classifying, summarizing, and ranking them, and then formulating a theme for the paper. Then, the literature is organized to explain the causal relationship between the various components. After that, the research framework and hypotheses were established and the questionnaire tool was designed, and IBM SPSS 22.0. and AMOS 21.0 for Windows (Armonk, NY, USA) were used to calculate and analyze the returned questionnaire samples and various information. Finally, the causal relationships were explored and inferred to obtain validated results, and then conclusions and recommendations were written, as shown in Figure 1.

According to the above research framework, there were five hypotheses in the study.

**Hypothesis 1 (H1).** *The level of involvement has a significant effect on familiarity.*

**Hypothesis 2 (H2).** *The level of involvement has a significant effect on destination imagery.*

**Hypothesis 3 (H3).** *Familiarity has a significant effect on travel motivation.*

**Hypothesis 4 (H4).** *Destination imagery has a significant effect on travel motivation.*

**Hypothesis 5 (H5).** *Travel motivation has a significant effect on travel intentions.*

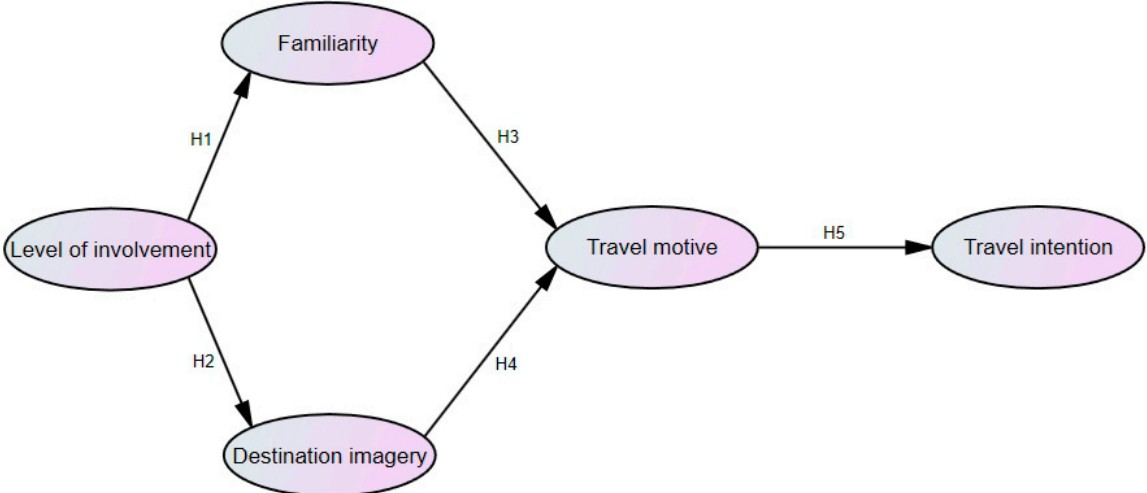

**Figure 1.** Research framework.

### 3.2. Research Tools and Scale Analysis

The purpose of this study was to analyze the impact of people's involvement in tourism programs on destination imagery, familiarity, motivation, and travel intentions. The study began in January 2018 with the initial development of the research questions and framework. After two years of careful evaluation and analysis, the study framework was refined and the questionnaire tool was designed in mid-2019. The questionnaire consisted of 3 parts. Part I provided background information on gender (male and female), age (under 19, 21–29, 30–39, 40–49, 50–59, 60+), length of residence or travel stay (short-term travel stay of 1 day, short-term travel stay of 1–3 days, short-term travel stay of more than 3 days, permanent or long-term residence of less than 1 year, permanent or long-term residence of 1–4 years, permanent or long-term residence of 5–8 years, permanent or long-term residence of 9–12 years, permanent or long-term residence of 13 years or more), etc. Part II referred to [57,58] to compile questions on the level of involvement in travel programs (3 questions in total). Part III referred to [59,60] to compile questions on destination imagery (15 questions in total). Part IV referred to [38,61,62] to compile travel familiarity questions (5 questions in total). Part V referred to [25,63,64] to compile travel motivation questions (9 questions in total). Part VI referred to [17,65–72] to compile the travel intention questions (3 questions in total). The research questionnaire design is shown in Table 1.

### 3.3. Research Limitations

The study was conducted on Taiwanese people, and the questionnaire was compiled with reference to the literature on travel program involvement [57,58], travel destination imagery [59,60], travel familiarity [38,61,62], travel motivation [25,63,64], and travel intention [17,65–72]. All subjects had no conflict of interest with the research team and agreed to provide data anonymously for analysis.

However, due to the constraints of research objectives, funding, manpower, and time, only 450 samples were collected for investigating the effects of destination imagery, familiarity, motivation, and travel intention. Any relevant research shortcomings or additional research questions will be provided as suggestions for subsequent research. It is hoped that subsequent researchers will continue the investigation and fill in the gaps.

**Table 1.** Study questionnaire design.

| Research Components | Questionnaire Content | Relevant Literature |
|---|---|---|
| Level of involvement | B1. I pay close attention when watching the travel program.<br>B2. The content of the travel program is interesting to me.<br>B3. I have relevant thoughts when I recall the content of the travel program. | [57,73] |
| Familiarity | A1. I am quite familiar with the travel program.<br>A2. I understand the content of the travel program.<br>A3. I often spend time collecting information about the travel program.<br>A4. I am more familiar with travel programs than the average person.<br>A5. I am more familiar with the travel program than my friends. | [61,70] |
| Destination imagery | C1. The tourism program introduces a variety of tourist contents, environment, and places.<br>C2. The tourism program introduces a variety of tourist attractions and activities.<br>C3. The tourism program introduces a wealth of natural tourist attractions.<br>C4. The tourist shopping facilities introduced by the tourism program are perfect.<br>C5. The tourist consultation service introduced by the tourism program is well developed.<br>C6. The tourist attractions introduced by the tourism programs are easily accessible by transportation.<br>C7. The tourist attractions introduced by the tourism program have a variety of accommodation options.<br>C8. The architecture of the tourist attractions introduced in the tourism program is unique.<br>C9. The tourist attractions introduced in the tourism program have a pleasant climate and are suitable for travel.<br>C10. The tourist attractions introduced by the tourism program have sound economic development.<br>C11. The security of the tourist attractions introduced by the tourism program is good, so that people can travel there with peace of mind.<br>C12. The tourist attractions introduced by the tourism program have distinctive customs and culture.<br>C13. The local residents of the tourist attractions introduced in the tourism program are friendly.<br>C14. The cost of the tourist attractions introduced in the tourism program is not high.<br>C15. The tourist attractions introduced in the tourism program have a special local atmosphere. | [71,72] |
| Travel motivation | EE1. Traveling allows you to eat unique food.<br>EE2. Traveling allows you to leave daily chores behind.<br>EE3. Traveling can relieve tired physical or mental conditions.<br>EE4. You can visit famous scenery and sightseeing spots in a perfect tourist environment.<br>EE5. It is natural to engage in travel. It allows me to participate in a variety of interesting activities.<br>EE6. I will go on a trip spontaneously in order to satisfy my shopping desires.<br>EE7. Because the travel cost is cheap.<br>EE8. I want to explore novel things.<br>EE9. I travel whenever I can to enhance the relationship between family members or friends. | [25,63,64] |
| Travel intent | D1. I am willing to travel to other countries for sightseeing.<br>D2. I am willing to visit other countries again.<br>D3. I am willing to recommend others to visit foreign countries. | [17,65,66] |

## 4. Results and Discussion

### 4.1. Demographic Analysis

The sample structure analysis of this study on respondents' involvement, familiarity, destination imagery, travel motivation, and travel intention showed that the majority of them were male, with a total of 309 persons, accounting for 68.7% of the total sample, and the rest were female, with a total of 141 persons, accounting for 31.3% of the total sample. The largest age group was 20–29 years old, with 350 persons, accounting for 77.8% of the total sample. In terms of education, the largest number of respondents were university students, with 344 persons, accounting for 76.4% of the total sample. For occupation, students were the most common, with 247 people, accounting for 54.9% of the total sample. For family status, the largest number of respondents were unmarried, 384, accounting for 85.3% of the total sample. Regarding the monthly income, the largest number of respondents (143) had a monthly income between $20,001 and $40,000, accounting for 31.8% of the total sample.

### 4.2. Reliability and Validity Analysis

#### 4.2.1. Examining the Offending Estimates

According to [74] and other scholars, the following three cases of estimation violations were considered in this study: (1) negative error variance, (2) standardized regression coefficient greater than 0.95, and (3) the error variance of the measurement was not significant. The results showed that the absolute values of the standardized regression coefficients for the level of involvement, familiarity, destination imagery, motivation, and travel intention ranged from 0.73 to 0.80, 0.51 to 0.88, 0.55 to 0.79, 0.46 to 0.81, and 0.78 to 0.92, respectively. None of them exceeded 0.95. The values of the error variances ranged from 0.01 to 0.06, with no negative error variances existing and they were significant. Therefore, the model does not contain any estimation violations and the measurement mode fitness test can be performed. The results are shown in Table 2.

**Table 2.** Examination of estimation violations.

| Item | Standardized Regression Coefficient | Error Variance |
|---|:---:|:---:|
| B1 ← Level of involvement | 0.73 | 0.03 |
| B2 ← Level of involvement | 0.80 | 0.04 |
| B3 ← Level of involvement | 0.79 | 0.04 |
| A1 ← Familiarity | 0.68 | 0.06 |
| A2 ← Familiarity | 0.85 | 0.04 |
| A3 ← Familiarity | 0.88 | 0.04 |
| A4 ← Familiarity | 0.73 | 0.05 |
| A5 ← Familiarity | 0.51 | 0.05 |
| C1 ← Destination imagery | 0.72 | 0.02 |
| C2 ← Destination imagery | 0.69 | 0.02 |
| C3 ←Destination imagery | 0.75 | 0.02 |
| C4 ← Destination imagery | 0.74 | 0.02 |
| C5 ← Destination imagery | 0.79 | 0.01 |
| C6 ← Destination imagery | 0.68 | 0.02 |
| C7 ← Destination imagery | 0.77 | 0.02 |
| C8 ← Destination imagery | 0.71 | 0.02 |
| C9 ← Destination imagery | 0.76 | 0.01 |
| C10 ← Destination imagery | 0.75 | 0.02 |
| C11 ← Destination imagery | 0.77 | 0.02 |
| C12 ← Destination imagery | 0.75 | 0.02 |
| C13 ← Destination imagery | 0.74 | 0.02 |
| C14 ← Destination imagery | 0.55 | 0.03 |
| C15 ← Destination imagery | 0.71 | 0.02 |
| EE1 ← Travel motive | 0.56 | 0.06 |
| EE2 ← Travel motive | 0.50 | 0.05 |

**Table 2.** *Cont.*

| Item | Standardized Regression Coefficient | Error Variance |
|---|---|---|
| EE3 ← Travel motive | 0.52 | 0.03 |
| EE4 ← Travel motive | 0.46 | 0.03 |
| EE5 ←Travel motive | 0.56 | 0.06 |
| EE6 ← Travel motive | 0.78 | 0.03 |
| EE7 ← Travel motive | 0.81 | 0.04 |
| EE8 ← Travel motive | 0.79 | 0.04 |
| EE9 ← Travel motive | 0.75 | 0.03 |
| D1 ← Travel intention | 0.78 | 0.01 |
| D2 ← Travel intention | 0.86 | 0.02 |
| D3 ←Travel intention | 0.92 | 0.02 |

### 4.2.2. Multivariate Assessment of Normality

In this study, the skewness values were all within 2, and the kurtosis values were all within the standard range of 7, i.e., the observed variables were all non-multivariate normality distributions according to the criteria of [75], as shown in Tables 3–7.

**Table 3.** Summary of the skewness and kurtosis of the observed variables—familiarity.

| Item (Variable) | Skew | C.R. | Kurtosis | C.R. |
|---|---|---|---|---|
| A1 | −0.24 | −2.06 | −0.88 | −3.82 |
| A2 | −0.26 | −2.06 | −0.78 | −3.39 |
| A3 | −0.04 | −0.31 | −0.90 | −3.90 |
| A4 | 0.37 | 3.21 | −0.56 | −2.42 |
| A5 | 0.03 | 0.25 | −0.08 | −0.36 |
| Multivariate | | | 14.76 | 18.72 |

**Table 4.** Summary of the skewness and kurtosis of the observed variables—level of involvement.

| Item (Variable) | Skew | C.R. | Kurtosis | C.R. |
|---|---|---|---|---|
| B1 | −0.82 | −7.09 | 1.12 | 4.87 |
| B2 | −0.10 | −0.88 | −0.28 | −1.21 |
| B3 | 0.01 | 0.08 | 0.07 | 0.31 |
| Multivariate | | | 6.19 | 12.00 |

**Table 5.** Summary of the skewness and kurtosis of the observed variables—destination imagery.

| Item (Variable) | Skew | C.R. | Kurtosis | C.R. |
|---|---|---|---|---|
| C1 | −0.44 | −3.77 | 0.90 | 3.88 |
| C2 | −0.59 | −5.08 | 1.34 | 5.82 |
| C3 | −0.50 | −4.33 | 0.84 | 3.64 |
| C4 | −0.15 | −1.26 | 0.21 | 0.91 |
| C5 | −0.22 | −1.88 | 0.49 | 2.13 |
| C6 | −0.47 | −4.08 | 0.79 | 3.41 |
| C7 | −0.08 | −0.71 | 0.01 | 0.05 |
| C8 | −0.35 | −3.00 | 0.70 | 3.03 |
| C9 | −0.45 | −3.92 | 0.82 | 3.56 |
| C10 | −0.02 | −0.19 | 0.17 | 0.72 |
| C11 | −0.29 | −2.52 | 0.25 | 1.09 |
| C12 | −0.37 | −3.23 | 0.61 | 2.63 |
| C13 | −0.36 | −3.15 | 0.68 | 2.94 |
| C14 | −0.12 | −1.03 | −0.26 | −1.11 |
| C15 | −0.34 | −2.95 | 0.78 | 3.39 |
| Multivariate | | | 126.73 | 59.52 |

**Table 6.** Summary of the skewness and kurtosis of the observed variables—travel motive.

| Item (Variable) | Skew | C.R. | Kurtosis | C.R. |
|---|---|---|---|---|
| EE1 | 0.17 | 1.44 | −0.64 | −2.75 |
| EE2 | 0.01 | 0.06 | −0.49 | −2.12 |
| EE3 | −0.93 | −8.06 | 1.29 | 5.58 |
| EE4 | −0.96 | −8.31 | 1.37 | 5.93 |
| EE5 | −0.39 | −3.39 | −0.53 | −2.30 |
| EE6 | −0.40 | −3.50 | −0.09 | −0.38 |
| EE7 | −0.31 | −2.64 | −0.51 | −2.22 |
| EE8 | −0.14 | −1.20 | −0.65 | −2.83 |
| EE9 | −0.26 | −2.25 | −0.10 | −0.43 |
| Multivariate | | | 28.28 | 21.31 |

**Table 7.** Summary of the skewness and kurtosis of the observed variables—travel intention.

| Item (Variable) | Skew | C.R. | Kurtosis | C.R. |
|---|---|---|---|---|
| D1 | −0.85 | −7.37 | 0.98 | −0.59 |
| D2 | −0.66 | −5.71 | 0.44 | 1.89 |
| D3 | −0.48 | −4.17 | −0.14 | 4.23 |
| Multivariate | | | 8.86 | 17.16 |

4.2.3. Confirmatory Factor Analysis

Reliability and Convergent Validity

This analysis was conducted to measure the convergent validity and the construct validity of the questionnaire using confirmatory factor analysis. The factor loadings in this study were based on the following criteria recommended by [76] to determine whether a question should be included in a factor study: factor loadings between 0.45 and 0.55 were considered fair, 0.55 to 0.63 were considered good, 0.63 to 0.71 were considered very good, and 0.71 or more were considered excellent. Therefore, the loadings of the factors in this study all met the criteria of [76]. After examining the statistical results, it was found that the MI values of A1 and A5 for familiarity, C1, C2, C4, C5, C6, C7, C8, and C14 for destination imagery, and EE1, EE2, EE3, EE4, EE5, and EE6 for travel motivation were too high, so it was decided to delete the above questions. The validity of the measurement model was examined by validation factor analysis to determine whether the variables of each measure converged to the potential variables of the desired measure. The average variances extracted between the potential variables and their corresponding measures were calculated as the average variances extracted for each observed variable, representing the average explanatory power of each observed variable for the potential variable. In this study, the average variances extracted were all above 0.5. This result met the criteria of [77], and therefore this study possessed convergent validity. The composition reliability of the potential variables was measured according to the recommendation of [77], and the higher the value, the higher the internal consistency of the measures and the more the construct validity of the potential variables could be measured. The results of this study showed that the component reliability values of all the constructs were above 0.6, which was consistent with the recommendation of [77], and therefore the internal quality of this study model was good.

The results of the reliability analysis and convergent validity of the confirmatory factor analysis are shown in Table 8, which shows that all three indicators of the validated analysis of this study, such as the factor loadings, the average variances extracted, and the construct reliability, met the standards [76,77].

**Table 8.** Summary of the convergent validity and construct reliability.

| Item | Standardized Factor Loading | Non-Standardized Factor Loading | S.E. | C.R. (t-Value) | p | SMC | C.R. | AVE |
|---|---|---|---|---|---|---|---|---|
| A2 ← Familiarity | 0.83 | 1.00 | | | | 0.70 | 0.86 | 0.67 |
| A3 ← Familiarity | 0.92 | 1.11 | 0.06 | 18.78 | *** | 0.84 | | |
| A4 ← Familiarity | 0.70 | 0.81 | 0.05 | 16.02 | *** | 0.49 | | |
| B1 ← Level of involvement | 0.73 | 1.00 | | | | 0.53 | 0.81 | 0.59 |
| B2 ← Level of involvement | 0.80 | 1.22 | 0.09 | 13.96 | *** | 0.65 | | |
| B3 ← Level of involvement | 0.79 | 1.19 | 0.09 | 13.94 | *** | 0.62 | | |
| C3 ← Destination imagery | 0.72 | 1.00 | | | | 0.52 | 0.90 | 0.58 |
| C9 ← Destination imagery | 0.77 | 1.05 | 0.07 | 15.88 | *** | 0.60 | | |
| C10 ← Destination imagery | 0.71 | 0.96 | 0.07 | 14.45 | *** | 0.50 | | |
| C11 ← Destination imagery | 0.82 | 0.15 | 0.07 | 16.55 | *** | 0.67 | | |
| C12 ← Destination imagery | 0.81 | 1.10 | 0.07 | 16.47 | *** | 0.66 | | |
| C13 ← Destination imagery | 0.78 | 1.06 | 0.07 | 15.96 | *** | 0.61 | | |
| C15 ← Destination imagery | 0.72 | 0.95 | 0.06 | 14.71 | *** | 0.52 | | |
| D1 ← Travel intent | 0.78 | 1.00 | | | | 0.61 | 0.89 | 0.73 |
| D2 ← Travel intent | 0.86 | 1.37 | 0.07 | 19.42 | *** | 0.73 | | |
| D3 ← Travel intent | 0.92 | 1.34 | 0.07 | 19.97 | *** | 0.84 | | |
| EE7 ← Travel motive | 0.90 | 1.00 | | | | 0.81 | 0.86 | 0.68 |
| EE8 ← Travel motive | 0.89 | 1.00 | 0.05 | 19.78 | *** | 0.79 | | |
| EE9 ← Travel motive | 0.67 | 0.65 | 0.04 | 15.39 | *** | 0.44 | | |

*** $p < 0.000$.

Discriminant Validity

The discriminant validity presents whether there is a significant relationship between two or more constructs, meaning whether it has good explanatory power [78]. In this analysis model, the 95% confidence interval of the correlation coefficient between the constructs was calculated using a bootstrap sampling assignment. If the number 1 does not appear in the coefficient 95% confidence interval, it means that the constructs have good discriminant validity [79–81]. The results in Table 9 show that the 95% confidence interval of the correlation coefficient of this construct bootstrap does not contain the number 1, so it means that there is good discriminant validity.

**Table 9.** Bootstrap correlation coefficients 95% confidence intervals.

| | | | Bias-Corrected | | | Percentile Method | |
|---|---|---|---|---|---|---|---|
| | | | Estimates | Lower Bound | Upper Bound | Lower Bound | Upper Bound |
| Level of involvement | ↔ | Familiarity | 0.62 | 0.54 | 0.71 | 0.54 | 0.70 |
| Level of involvement | ↔ | Destination imagery | 0.57 | 0.47 | 0.66 | 0.47 | 0.66 |
| Level of involvement | ↔ | Travel motive | 0.39 | 0.27 | 0.50 | 0.27 | 0.50 |
| Level of involvement | ↔ | Travel intention | 0.53 | 0.43 | 0.61 | 0.43 | 0.61 |
| Familiarity | ↔ | Destination imagery | 0.37 | 0.27 | 0.47 | 0.27 | 0.47 |
| Familiarity | ↔ | Travel motive | 0.27 | 0.16 | 0.38 | 0.15 | 0.38 |
| Familiarity | ↔ | Travel intention | 0.27 | 0.17 | 0.36 | 0.17 | 0.36 |
| Travel motive | ↔ | Destination imagery | 0.33 | 0.21 | 0.43 | 0.20 | 0.43 |
| Travel intention | ↔ | Destination imagery | 0.73 | 0.63 | 0.81 | 0.63 | 0.81 |
| Travel motive | ↔ | Travel intention | 0.41 | 0.31 | 0.51 | 0.30 | 0.50 |

Overall Structural Model Analysis

The present structural model analysis was performed with reference to [74,81–83], and the overall model fitness was evaluated by seven indicators, including $\chi^2$ test, the ratio of $\chi^2$ to degrees of freedom, GFI, AGFI, RMSEA, CFI, and PCFI. As shown in Table 10, the corrected ratio of $\chi^2$ to degrees of freedom was 3.61 (less than the recommended value of 3), the value of GFI was 0. 90 (equal to 0.90), the value of AGFI was 0.90 (greater than 0.80), the value of RMSEA was 0.07 (less than 0.08), the value of CFI was 0.92 (greater than 0.90), and the value of PCFI was 0.79 (greater than 0.50). Thus, these results indicated that the model was acceptable (Figure 2).

**Table 10.** Summary of study hypotheses and validation results.

| Study Hypothesis | Path Coefficient | Validation Result |
| --- | --- | --- |
| Hypothesis H1: The level of involvement has a significant effect on familiarity. | 0.63 | Valid |
| Hypothesis H2: The level of involvement has a significant effect on destination imagery. | 0.58 | Valid |
| Hypothesis H3: Familiarity has a significant effect on travel motivation. | 0.18 | Valid |
| Hypothesis H4: Destination imagery has a significant effect on travel motivation. | 0.31 | Valid |
| Hypothesis H5: Travel motivation has a significant effect on travel intentions. | 0.44 | Valid |

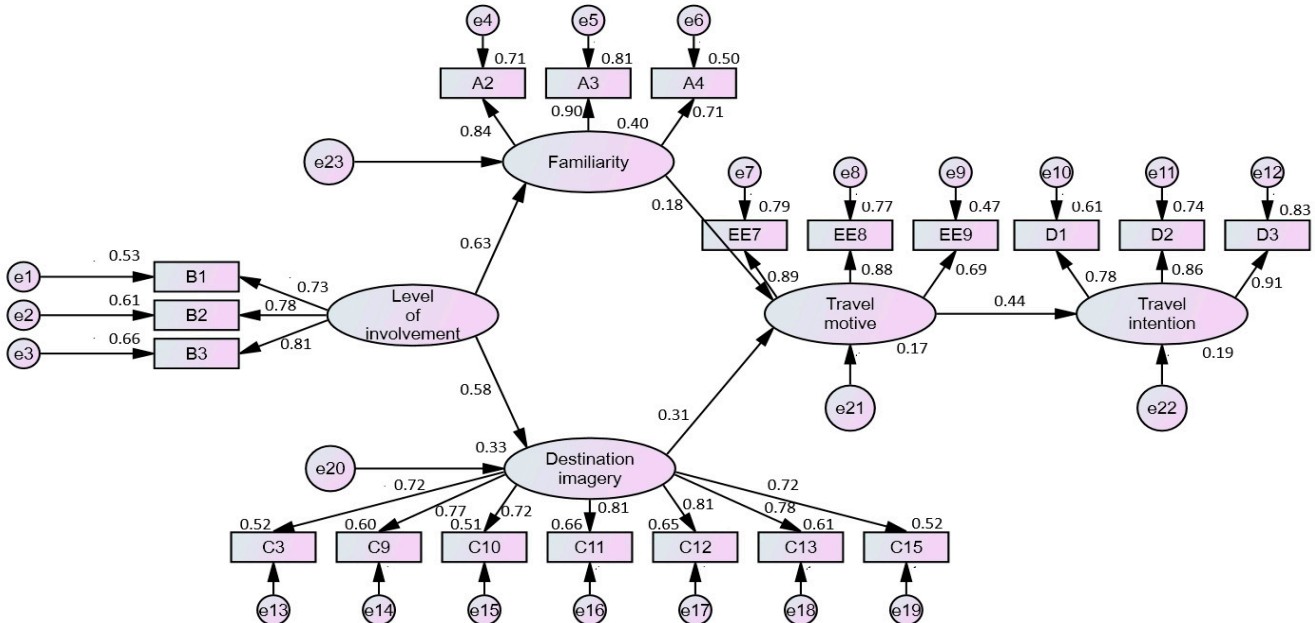

**Figure 2.** The study model of the effect of travel program involvement on familiarity, destination imagery, travel motivation, and travel intention.

a.   The relationship between the level of involvement and familiarity

As noted by [44] and [45], when people receive correct travel information through learning or experience, a high degree of familiarity with the tourist area is generated. The deeper the understanding and knowledge of the tourist area, the higher the familiarity, and the higher the positive perception of the tourist area. In this study, the higher the degree of involvement, the higher the familiarity, which is consistent with [19].

b.   The relationship between the level of involvement and destination imagery

In literature [41–43], it is stated that the more frequently people receive information, the more impressed they are with the local tourism. It is clear that the degree of access to tourism information affects people's perceptions and impressions of tourist areas. Good destination imagery facilitates tourists' travel planning. According to the results of this study, the level of involvement in watching tourism programs had a significant effect on destination imagery, meaning that the higher the level of involvement of the viewer, the higher the destination imagery, which is comparable to the findings of [39,40].

c.   The relationship between familiarity and travel motive

Proper tourism planning requires that individuals know the tourist places themselves. Travel programs can stimulate or change the viewers' perceptions of tourist places, enhance the resources such as scenery, ecology, products, and activities of tourist places, and increase people's intention to travel [50,51]. Therefore, it is believed that the travel motive of those who watch travel programs will be influenced by familiarity.

d. The relationship between destination imagery and travel motive

The more people know about a tourist destination, the quicker they can plan for future travel [49]. Therefore, the deeper the involvement in tourism programs, the more opportunities there are to obtain more and better tourism information. The higher the positive destination imagery, the better the destination imagery will be for the public.

e. The relationship between travel motive and travel intention

Travel intention is an assessment of the multiple factors that influence travel within an individual and determine whether people will perform travel behaviors [53,54]. Therefore, past travel experiences or acquired travel information can influence people's perceptions of travel places and further constitute travel attitudes and induce travel motivation.

*4.3. The Relationship between Tourism Programs, Tourists, and Rural Sustainable Development*

According to the results of our study, the degree of involvement in tourism programs is associated with rural tourism imagery, familiarity, motivation, and willingness. The investigators concluded that although tourism development has become a mature industry over a long period of time [60,61], due to its competitiveness, the high similarity of tourism characteristics in neighboring countries [29], and the impact of the COVID-19 epidemic, it has become the most important trend to create a brand image, increase tourists' attention to the scenic areas, and enhance travel ideas and motivation to facilitate the achievement of travel behavior [30].

Still, due to the advancement of technology, media has become involved in people's daily life [19], and the use of media communication platforms to assist in the marketing of scenic spots can indeed contribute to deepening tourists' attention to them [84]. As a result, when people gain travel knowledge or experience of a scenic area through tourism programs, a deep impression has already been made in tourists' minds.

Moreover, when tourism programs present tourist information of scenic spots, the video editor's editing, the program producer's integrated planning, and the on-set host's explanation can accelerate tourists' knowledge of local tourism resources [85].

Furthermore, tourism activities must first stimulate tourists' imagination of tourist destinations to generate tourism attraction [51], and then produce tourism motivation [21,30], in order to stimulate people's desire to complete tourism motivation and realize their dreams of tourism planning.

Therefore, the packaging and production of travel programs can enhance people's understanding of and involvement in the programs. Increasing the level of involvement in travel programs can indirectly influence tourists' rural travel imagery, increase the degree of knowledge of tourism resources in scenic areas, promote tourism motivation, and achieve tourism aspirations.

Therefore, we believe that the level of tourists' involvement in travel programs can indeed help rural tourism development achieve sustainable development goals.

## 5. Conclusions and Suggestions

*5.1. Conclusions*

### 5.1.1. The Level of Involvement of the Viewer Has a Significant Effect on Familiarity

There is a significant correlation between the level of involvement and familiarity, indicating that there is a mutual influence between the level of involvement of viewers in travel programs and familiarity. From the component path analysis, it was found that the "level of involvement" component had a significant effect on the dependent variable "familiarity". The higher the viewers' involvement in the travel program, the higher their familiarity with the destination represented by the video, which in turn stimulates them to collect more information, thus affecting their familiarity with the destination.

### 5.1.2. Viewers' Levels of Involvement Have a Significant Effect on Destination Imagery

There is a significant correlation between the level of involvement and destination imagery, indicating that there is a reciprocal relationship between viewers' involvement in the tourism program and destination imagery. From the component path analysis, it was found that the "level of involvement" component had a significant effect on the dependent variable "destination imagery". In other words, the stronger the viewer's involvement in the content of the travel program, the more positive the image of the destination area represented.

### 5.1.3. Familiarity of Viewers Has a Significant Effect on Travel Motivation

There is a significant correlation between familiarity and motivation to travel, indicating that viewers' familiarity with tourism programs and motivation to travel are mutually influential. The results of the study indicate that the more familiar the viewer is with the tourism resources of the region, the greater the impact on the viewer's willingness to spend money on travel.

### 5.1.4. Viewers' Destination Imagery Has a Significant Effect on Travel Motivation

There is a significant correlation between destination imagery and motivation to travel, indicating that viewers' destination imagery from tourism programs and motivation to travel are mutually influential. The results of the study show that the more viewers identify with the landscape and environmental imagery, the more likely they are to be motivated to visit the destination by the features introduced in the program, suggesting that in order to attract people to a destination, it is important to establish a high level of awareness and destination imagery.

### 5.1.5. Viewers' Motivation to Travel Has a Significant Impact on Travel Intentions

There is a significant correlation between travel motivation and travel intention, indicating that viewers' travel motivation and travel intention regarding travel programs are mutually influential. The results of the study show that when viewers travel to different countries, they are mostly motivated or driven to travel by programs on location.

### *5.2. Recommendations for Future Research*

5.2.1. Recommendations for Program Operators and Governments on Local Tourism Marketing Practices

(1) This study confirms that constructing destination imagery through tourism programs makes a significant contribution to increasing travel intentions. Therefore tourism marketers can cooperate with tourism program operators and utilize media such as TV, movies, cell phones, and YouTubers to integrate special attractions through product placement marketing. With famous program hosts, local culture and cuisine can be conveyed while local scenery can be engraved in the viewer's mind, enhancing the visitor's imagery of the destination and providing a real boost to tourism marketing. By presenting the unique local food and cultural experiences in a package, visitors can also gain a deeper understanding of culture and food through the experience.

(2) The study found that the higher the viewer's involvement in the content of a travel program, the better the familiarity with the destination. The level of viewer involvement positively affects the destination imagery, which means that the most important motivation for travel is the level of involvement in the program. Thus, the motivation of travelers to visit a destination is based on the attractiveness of the content of the program. It is recommended that the government agencies should develop strategies to increase viewers' involvement in tourism programs. On the other hand, many factors influence the level of audience involvement, including the role, style, and popularity of the host. Therefore, if governments at all levels want to use tourism programs to market local attractions in the future, they should advise the hosts and filming locations to properly integrate local culture and scenery into the program

content. The program's content will certainly increase the level of involvement of travelers and trigger their intention to travel abroad.

(3)　Our findings show that the higher the level of involvement in tourism programs, the higher the desire and motivation to travel, and the more the viewers perceive the tourist area as a charming city. Therefore, tourism programs can be used to help create a different experience for visitors. The local community can make use of its own unique resources, culture, and cuisine to develop different thematic areas for different types of visitors. Local governments can utilize their own cultural and gastronomic and rural tourism-specific resources to create charming cities with special features to meet the needs of different tourists. In addition, the familiarity of viewers with tourism programs has a significant impact on tourism motivation. With the rise of internet celebrities and live streaming channels, hiring famous YouTubers to introduce local sightseeing spots, local food and strong ethnic culture through live streaming can increase the viewing frequency and attract more tourists to visit the rural tourism.

### 5.2.2. Study Limitations

1.　Due to financial and human resources constraints, this study was conducted on Taiwan only, so the inference of the results is limited, and subsequent studies may be extended to other regions or countries for verification.

2.　This study was conducted only on the basis of factors such as level of involvement, familiarity, destination imagery, travel motivation, and travel intention. Since there are many influencing factors, an intervention or moderated variable analysis can be conducted to facilitate subsequent studies to understand more deeply whether familiarity and destination imagery affect travel motivation.

3.　This study was conducted in a quantitative manner using questionnaires, and it was difficult to control the situation at the time of completing the questionnaires. Subsequent related studies can be conducted by qualitative research using the interview method or action research, with case-specific observations, in order to compensate for the lack of quantification.

4.　For the convenience of questionnaire distribution and collection, this study adopted a convenience sampling method and did not classify the sample size into categories, so the degree of difference was not significant. Subsequent studies may conduct questionnaire surveys in a stratified sampling mode to explore the correlations and differences.

**Author Contributions:** Project administration, M.-Y.Z.; data curation, funding acquisition, supervision, C.-C.C.; validation, visualization, writing—original draft preparation, H.-H.L.; funding acquisition, investigation, methodology, writing—original draft preparation, C.-H.T.; Resources, writing—review and editing, C.-H.H. All authors have read and agreed to the published version of the manuscript.

**Funding:** This research received no external funding.

**Data Availability Statement:** No data support.

**Acknowledgments:** We thank everyone who helped this manuscript.

**Conflicts of Interest:** The authors declare no conflict of interest.

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
