# Peer review of "Research on the Impact of Popular Tourism Program Involvement on Rural Tourism Image, Familiarity, Motivation and Willingness"

_sustainability, doi:10.3390/su13094906_

Round 1
Reviewer 1 Report
The work has a correct structure of a scientific article, the results of the survey are elaborated using statistical methods and discussed. However, despite the high evaluation of the work it is advisable to apply the following improvements that will increase quality of the work:
- The title of the work needs revision. It uses the phrase 'rural tourism destination' (line 3), while the work lacks references to rural areas.
- The abstract uses the phrase 'the motivation to travel in the last village' (line 21), while the paper has no reference to villages. The authors do not write clearly which area the study concerned. The conclusion (lines 528-529) implies that the study was conducted in Taiwan, but this was not mentioned before.
- Key words need to be verified. The last two are in my opinion redundant; it would be better to use the terms 'experience tourism' and Taiwan.
- The Introduction chapter should be enriched with references to experience tourism, experiences in tourism and the relationship of the research to sustainable tourism, sustainable development.
- The Methodology chapter should be supplemented with information on how the surveys were conducted, how respondents were selected and whether the surveys complied with ethical requirements? Please provide details of the Research Ethics Committee that approved this study, or add "Ethical review and consent were waived for this study for reason (please provide detailed justification)". In addition, it is necessary to state which area the study concerned, together with a location map of that area.
- In Table 1, it is advisable to add symbols B1-D1, which the Authors use in Tables 2-8 and in Figure 2.
- In figure 2 the Authors use symbols e1-e20. However, it is difficult to find out what they mean. It is advisable to explain them.
- It is proposed to broaden the discussion.
- In the conclusions (line 516-525) the authors refer several times to cities. However, in the title there is a reference to rural tourism. Is there not a contradiction here?
- It is advisable to revise the entire text to eliminate repetitions.
Author Response
Reviewers 1
The work has a correct structure of a scientific article, the results of the survey are elaborated using statistical methods and discussed. However, despite the high evaluation of the work it is advisable to apply the following improvements that will increase quality of the work:
- The title of the work needs It uses the phrase 'rural tourism destination' (line 3), while the work lacks references to rural areas.
Thank you reviewer
We will make adjustments. “ The impact of tourism program involvement on the imagery and familiarity of rural tourism destinations, as well as the
motiva-tion and desire to travel - A Case Study in Taiwan “
- The abstract uses the phrase 'the motivation to travel in the last village' (line 21), while the paper has no reference to The authors do not write clearly which area the study concerned. The conclusion (lines 528-529) implies that the study was conducted in Taiwan, but this was not mentioned before.
Thank you reviewer
People carry out recreational and tourism activities through rural tourism to obtain the leisure benefits of relaxation, entertainment and recovery of the body and mind; rural tourism, through the development of the leisure
tourism industry, flourishes the transformation of the local economy and society. Therefore, rural tourism is of great importance to the urban-rural interaction between the people. Taiwan has been mentioned in the title.
- Key words need to be The last two are in my opinion redundant; it would be better to use the terms 'experience tourism' and Taiwan.
Thank you reviewer
We have changed "environment, tourism marketing" to "experience tourism, Taiwan"
- The Introduction chapter should be enriched with references to experience tourism, experiences in tourism and the relationship of the research to sustainable tourism, sustainable
Thank you reviewer
We have added relevant narratives. Such as line 78-80, 98-101.
- The Methodology chapter should be supplemented with information on how the surveys were conducted, how respondents were selected and whether the surveys complied with ethical requirements? Please provide details of the Research Ethics Committee that approved this study, or add "Ethical review and consent were
waived for this study for reason (please provide detailed justification)". In addition, it is necessary to state which area the study concerned, together with a location map of that area.
Thank you reviewer
In this study, the questionnaire was distributed by intentional sampling, and a representative sample was selected according to the judgment of experts.
- In Table 1, it is advisable to add symbols B1-D1, which the Authors use in Tables 2-8 and in Figure
Thank you reviewer
The data has been added. See Table 2-8 and Figure 2.
- In figure 2 the Authors use symbols e1-e20. However, it is difficult to find out what they mean. It is advisable to explain
Thank you reviewer
e1-e20 represent structural residuals, measurement residuals, errors, and unexplainable variations; in SEM, each variable must have a name, and cannot be blank, otherwise it cannot be analyzed.
- It is proposed to broaden the
Thank you reviewer
We only discuss the path relationship of tourism programs involved in the image and familiarity of rural tourism destinations, as well as tourism
motivation and desire. For other topics to be discussed, they will become the recommendations of this research, and we look forward to further research and discussion by future generations.
- In the conclusions (line 516-525) the authors refer several times to However, in the title there is a reference to rural tourism. Is there not a contradiction here?
Thank you reviewer
We changed the city to rural tourism.
- It is advisable to revise the entire text to eliminate
Thank you reviewer
We have moderate adjustments.

Reviewer 2 Report
The paper has a quite clear research goal: to acknowledge whether viewers involvement in tourists programs has a connection with the factors of familiarity, imagery, motivation and intention to travel.
Starting with the research methodology, I am not very satisfied with the way the authors justified the built validity of the questionnaire. It would be quite helpful not only to cite the source of the questionnaire but moreover to cite the source of each item (question). I had to go through- almost- the whole list of referenced papers to understand the use and source of each question. In a sentence, it would be extremely helpful in table 1 to add a reference column indicating the source of each question (item).
Regarding the analysis, it was quite detailed, learning no space for misunderstandings. The model seems to have an appropriate fit, the numbers are all intact.
The conclusions are also well written and understandable, but I am not sure that I agree with the conclusion that "tourism programs can be used to help create a different experience for visitors". The initial research question was to investigate the involvement (as a latent variable) in correlation with motivation and intention to travel. Therefore, I would suggest reforming the conclusion towards the initial research goals.
In thick lines, the paper underpins that viewers of tourism programs are motivated and more willing to travel there. The contribution to knowledge and research is not extremely significant. As a fellow scientist, I think that the contribution of the paper is the intention to acknowledge the extent of 3D or 7D (or even hologrammatic) tourist program will enhance the image and the intention to travel.
I think that the paper could be published but to be more accurate with the methodology I would strongly suggest reforming table 1, adding the reference column.
Author Response
Reviewers 2
Comments and Suggestions for Authors
The paper has a quite clear research goal: to acknowledge whether viewers involvement in tourists programs has a connection with the factors of familiarity, imagery, motivation and intention to travel.
Starting with the research methodology, I am not very satisfied with the way the authors justified the built validity of the questionnaire. It would be quite helpful not only to cite the source of the questionnaire but moreover to cite the source of each item (question). I had to go through- almost- the whole list of referenced papers to understand the use and source of each question. In a sentence, it would be extremely helpful in table 1 to add a reference column indicating the source of each question (item).
Regarding the analysis, it was quite detailed, learning no space for misunderstandings. The model seems to have an appropriate fit, the numbers are all intact.
The conclusions are also well written and understandable, but I am not sure that I agree with the conclusion that "tourism programs can be used to help create a different experience for visitors". The initial research question was to investigate the involvement (as a latent variable) in correlation with motivation and intention to travel. Therefore, I would suggest reforming the conclusion towards the initial research goals.
In thick lines, the paper underpins that viewers of tourism programs are motivated and more willing to travel there. The contribution to knowledge and research is not extremely significant. As a fellow scientist, I think that the contribution of the paper is the intention to acknowledge the extent of 3D or 7D (or even hologrammatic) tourist program will enhance the image and the intention to travel.
I think that the paper could be published but to be more accurate with the methodology I would strongly suggest reforming table 1, adding the reference column.
Thank you reviewer
We have added the source of the question Degree of involvement: Schiffman, Kanuk (2000); Wang, et Al., (2017) Familiarity: Baloglu (2001); Prentice (2004) Destination image: Llodrà et al., (2015); Chaulagain et al., (2019) Travel motivation: Kim et al., (2019), Heslin et al., (2020), Egger et al., (2020) Travel intention: Woodside &
Lysonski (1989); Fakeye & Crompton (1991); Lam & Hsu (2006)

Reviewer 3 Report
Thank you for the opportunity of reading and reviewing your interesting article. It addresses a topic which is interesting and actual. The literature is solid, however some issues can be addressed as well. I suggest several issues/references: social media role: Sharmin et al https://doi.org/10.3390/su13042308, web sites: Hu et al https://doi.org/10.1177/1096348005276496, cross-border context: https://doi.org/10.4335/12.3.349-371(2014)
The methodology is rather classical but adequate. The research is well conducted, and the results are well presented. Limitations, further research suggestions are also addressed.
I suggest authors to add a section or some paragraphs to address implications of their findings, both theoretical and practical.
Several tables could be added in some appendix, to smooth the experience of readers.
Most important, the title should be changed, now it is too long and not clear.
Good luck!
https://doi.org/10.4335/12.3.349-371
Author Response
Reviewers 3
Thank you for the opportunity of reading and reviewing your interesting article. It addresses a topic which is interesting and actual. The literature is solid, however some issues can be addressed as well. I suggest several issues/references: social media role:Sharmin et al https://doi.org/10.3390/su13042308, web sites:Hu et al https://doi.org/10.1177/1096348005276496, cross-border context:https://doi.org/10.4335/12.3.349-371(2014)
The methodology is rather classical but adequate. The research is well conducted, and the results are well presented. Limitations, further research suggestions are also addressed.
I suggest authors to add a section or some paragraphs to address implications of their findings, both theoretical and practical.
Several tables could be added in some appendix, to smooth the experience of readers. Most important, the title should be changed, now it is too long and not clear.
Good luck!
Thank you reviewer
Without affecting the structure, the topic has been streamlined. We have a description in line 154-172.

Round 2
Reviewer 3 Report
Thank you for providing a new version of your manuscript. Although some changes appear to be made, my comments have not been addressed so far. The authors claim about addressing them on lines 154-172, but my comments were related to final sections, at most.
Another suggestion of mine was about changing the title. The tile has been changed but I do not think it is more appropriate now, on the contrary.
Good luck!
Author Response
Review 3 -r2
thank you for providing a new version of your manuscript. Although some changes appear to be made, my comments have not been addressed so far. The authors claim about addressing them on lines 154-172, but my comments were related to final sections, at most.
Another suggestion of mine was about changing the title. The tile has been changed but I do not think it is more appropriate now, on the contrary.
Good luck!
Dear reviewer
We have added your suggestion.
Please refer to line 470-495.
Thank you very much for your assistance to make the manuscript more perfect.

Round 3
Reviewer 3 Report
Thank you for this new version. My comments and suggestions were addressed. I suggest small adjustment of the title, to be more attractive and citable.